# Comparison of "Orthodontic First" and "Surgery First" Approaches to Quality of Life in Orthognathic Surgery Patients: A Prospective Cohort Study

Chonakan Thitiyuk [1], Siripatra Patchanee [2,3], Awiruth Klaisiri [4] and Narissaporn Chaiprakit [1,3,*]

1 Division of Oral and Maxillofacial Surgery, Faculty of Dentistry, Thammasat University, Bangkok 12120, Pathum Thani, Thailand
2 Division of Orthodontics, Faculty of Dentistry, Thammasat University, Bangkok 12120, Pathum Thani, Thailand
3 Thammasat University Research Unit in Mineralized Tissue Reconstruction, Faculty of Dentistry, Thammasat University, Bangkok 12120, Pathum Thani, Thailand
4 Division of Restorative Dentistry, Faculty of Dentistry, Thammasat University, Bangkok 12120, Pathum Thani, Thailand
* Correspondence: nchaiprakit@gmail.com; Tel.: +66-2986-9213 (ext. 7155)

**Abstract:** This prospective cohort study aimed to assess changes in quality of life (QoL) for "orthodontic first" approach (OFA) and "surgery first" approach (SFA) patients. Sixty patients who underwent orthognathic surgery via either the OFA ($n = 30$) or the SFA ($n = 30$) provided self-administered questionnaires (SAQs) about their orthognathic quality of life (Thai version OQLQ). Data were collected at four time points: before surgery (T1), and postoperatively, at 1 (T2), 3 (T3) and 6 months (T4). Intragroup and intergroup comparisons were performed by the Friedman test and the Mann–Whitney U test, respectively. Both the OFA and the SFA showed that QoL scores gradually improved in each domain for 6 months after surgery. The pattern of improvement after surgery in the SFA group was: facial esthetics (E) at 1 month; awareness of facial deformities (A) and social aspects of deformity (S) at 3 months; oral function (F) at 6 months. The pattern of improvement after surgery in the OFA group was: facial esthetics (E) and social aspects of deformity (S) at 1 month; awareness of facial deformities (A) at 3 months; oral function (F) at 6 months. The most concerning domain for Thai patients in our center OQLQ was the facial esthetic domain.

**Keywords:** orthognathic surgery; dentofacial deformities; orthodontics; quality of life; surveys and questionnaires

## 1. Introduction

Dentofacial deformities are disharmonies between the facial, dental and bone structures that could impact a patient's quality of life (QoL), functionally and esthetically [1]. Orthognathic surgery combined with orthodontic treatment improves facial profile and occlusal function [2]. Orthognathic surgery also has significant psychosocial benefits for patients, in terms of self-confidence, relationships and social interaction [3]. Improvement of function, esthetic, and psychological status after orthognathic surgery increases the patient's QoL [4–6]. The "orthodontic first" approach (OFA) consists of a presurgical orthodontic treatment to decompensate teeth, causing worsened occlusion before surgery [7]: the study found that this approach is time-consuming and, during the presurgical orthodontic phase, deteriorates the facial profile [8,9]. The study also discovered that the presurgical orthodontic phase significantly worsened oral health-related quality of life (OHRQL) in the facial esthetics and oral function domains [9]. The "surgery first" approach (SFA) is an alternative option for orthognathic surgery, in some cases, that eliminates the drawbacks of the presurgical orthodontic phase, and utilizes the potency of the regional accelerated phenomenon (RAP), to facilitate postsurgical orthodontic tooth movement [10,11].

The treatment outcomes of orthognathic surgery can be evaluated physically by clinical and radiographic findings. Many studies have monitored the outcomes of the SFA, in terms of stability or relapse: the SFA had stability and relapse rates similar to the OFA [12–15]. In addition to stability, patient satisfaction and quality of life are important factors that medical teams should carefully address. Orthognathic surgery is reported to have significantly improved patient QoL [1,4,6,16,17].

Many tools have been used to assess QoL. WHOQoL or SF-36 is a generic health questionnaire. The Oral Health Impact Profile (OHIP) and Oral Impacts on Daily Performances (OIDP) have been commonly used, in regard to focusing on the oral health of patients. In addition, there is a set of questions, which pertains specifically to this patient group's condition, called the Orthognathic Quality of Life Questionnaire (OQLQ). The OQLQ was developed by Cunningham et al. in 2000, and consists of 22 items from four domains: facial esthetics (E), oral function (F), awareness of facial deformities (A) and social aspects of deformity (S). The patients rate accordingly, for each question, the degree of severity influencing their routine life [18]. The total score is rated from 0 to 88 points: the higher scores represent a greater influence on the patient's quality of life, while the lower scores represent a lower influence on the patient's quality of life. The studies showed that OQLQ is a suitable tool for assessing QoL, in particular for orthognathic surgery patients, when compared to generic health or oral health questionnaires [19,20]. The original English version has been translated and used in various countries, such as Brazil, Japan, South Korea, the United Kingdom and Italy. The results of OQLQs show that the questionnaire is used for the validity and reliability of its QoL assessment [2,9,21–23]. In Thailand, no standard tool for QoL assessment had been used until the OQLQ was translated and validated into the Thai version, in 2021. The Thai version of the OQLQ was used in this study, to investigate patients' QoL after orthognathic surgery, and to compare the differences in QoL changes between the OFA and the SFA.

## 2. Materials and Methods

### 2.1. Study Design

This prospective cohort study recruited 60 Thai patients who had dentofacial deformities, and who underwent orthognathic surgery (OFAs; $n = 30$, SFAs; $n = 30$) at the oral and maxillofacial clinic, Thammasat university hospital, between January and June 2021. A sample size was calculated by G-Power software, using data from a similar previous study. Power was conducted at 95%, and an increase of 10% in the case of some samples was lost to follow-up.

The patients underwent orthognathic surgery at the oral and maxillofacial clinic. The novel modification of the Low Z-plasty (NM-Low Z) technique was performed in mandibular osteotomy in this study [24]. There was only one surgeon, who performed all the orthognathic operations. The study was conducted according to the guidelines of the Declaration of Helsinki, and was approved by the Institutional Review Board of Thammasat university (RB/EC120/2563). Informed consent was obtained from all subjects involved in the study.

Intermaxillary wiring fixation (IMF) was used for 1 week at our center, following which, the patients were advised to consume a soft diet, and to perform open mouth exercise for 1 week. At 2 weeks postoperatively, early mobilization was started, and the patients were asked to begin to function.

### 2.2. Inclusion and Exclusion Criteria

Patients who had growth completion, and a diagnosis of skeletal type II and III, were included in this study. The participants had a complete series of identifiable lateral cephalograms. Growth completion was confirmed by a cervical vertebral maturation status of CS6 [25]. Patients who presented with cleft lip and/or palate, temporomandibular joint dysfunction, previous orthognathic surgery and serious underlying systemic disorders and psychological disorders, were excluded from this study.

The total 60 patients who were eligible for this trial, following the inclusion and exclusion criteria, were divided into two groups, according to orthodontic treatment approach (OFA or SFA). The SFA group included patients who had not received orthodontic treatment before surgery. The OFA group included patients who had presurgical orthodontic decompensation before orthognathic surgery. Demographic data and cephalometric data were collected from all the patients, for further analysis. The cephalometric data of the two groups, from lateral cephalograms before surgery, were compared, and showed no significant differences.

### 2.3. Data Collection

In the present study, the Thai version of the OQLQ was used for the QoL assessment of the patients. The patients were asked to complete questionnaires at four time points: before surgery (T1), and postoperatively, at 1 (T2), 3 (T3) and 6 months (T4).

The OQLQ consisted of 22 items, and was divided into four domains: facial esthetics (E; question 1, 7, 10, 11, 14), oral function (F; question 2–6), awareness of facial deformities (A; question 8, 9, 12, 13) and social aspects of deformity (S; question 15–22). The patients gave their score according to the severity of the impact each item had on them: a score of 4 meant that the item had impacted the patient a lot; a score of 1 meant that the item had had little impact on the patient; "N/A" indicated that the item had had no impact at all on the patient. A higher score signified a poorer QoL; a lower score signified a better QoL [18].

### 2.4. Data Analysis

The demographic data were compared, by using Pearson's chi-squared test for gender, and an independent T-test for age. The cephalometric data were compared by using an independent T-test. The mean scores of each domain and the total OQLQ were calculated. The Friedman test was used to compare means at each time point (T1, T2, T3 and T4) for intragroup comparison. The mean differences before surgery, and 1, 3 and 6 months after surgery (T1–T2, T1–T3 and T1–T4) were calculated and analyzed, using the Mann–Whitney U test for intergroup comparisons. The data were analyzed, using SPSS version 22 for Mac (Armonk, NY, USA, IBM Corp.) To compare the effect sizes (Cohen's d) of changes in scores between groups, the effect sizes of each group were calculated, by using the mean score at 6 months after surgery (M2), minus the mean score before surgery (M1), and were then divided by pooled standard deviation (SD) (Cohen's d = $(\frac{M2-M1}{pooled\ SD})$ [26].

### 3. Results

The demographic data for the OFA and the SFA patients were compared (Table 1). There were no significant differences in sex or age between the groups. The patients in this study consisted of skeletal II and III. ANB was used for comparison in the horizontal dimension, and SN-MP was used for comparison in the vertical dimension. The results showed no significant difference in cephalometric data, for both the horizontal and the vertical skeletal measurements, between the two groups at the time before surgery.

The average OQLQ score before surgery (T1) was 53.50 ± 15.96 points in the OFA group, and 51.90 ± 15.27 points in the SFA group. The total average score was 52.7 points, which was 59.89% of the total 88 points.

The OQLQ scores at each time point were compared (Tables 2 and 3): at 1 month after surgery, the facial esthetics and social aspects of the deformity domains improved significantly in the OFA group, whereas in the SFA group, only the facial esthetics domain improved significantly; at 3 months after surgery, the social aspects of deformity, facial esthetics, oral function and total score improved significantly in the OFA group, while in the SFA group, the social aspects of deformity, facial esthetics, awareness of facial deformities and total score improved significantly; at 6 months after surgery, all domains and total scores improved significantly.

**Table 1.** The demographic data were compared by using Pearson's chi-squared test ([†]) for gender, and the independent T-test ([‡]) for age. The cephalometric data were compared by using the independent T-test ([‡]). The duration of the presurgical orthodontic phase is also exhibited in the table.

| | OFA | SFA | Total | *p* Value |
|---|---|---|---|---|
| **Gender (*n*, %)** | | | | |
| Male | 14 (46.7%) | 11 (36.7%) | 25 (41.7%) | 0.432 [†] |
| Female | 16 (53.3%) | 19 (63.3%) | 35 (58.3%) | (Pearson's chi-squared Test) |
| **Age (years)** | | | | |
| Mean ± SD | 25.67 ± 4.88 | 24.67 ± 5.892 | 25.06 ± 3.774 | 0.477 [‡] |
| | | | | (Independent T-Test) |
| **Type of deformity** | | | | |
| Skeletal II (*n*, %) | 5 (16.7%) | 3 (10.0%) | 8 (13.3%) | |
| ANB (degrees) | 4.52 | 8.50 | 6.01 | 0.199 [‡] |
| | | | | (Independent T-Test) |
| SN-MP (degrees) | 35.20 | 48.77 | 40.29 | 0.073 [‡] |
| | | | | (Independent T-Test) |
| Skeletal III (*n*, %) | 25 (83.3%) | 27 (90.0%) | 52 (86.7%) | |
| ANB (degrees) | −4.50 | −3.19 | −3.90 | 0.099 [‡] |
| | | | | (Independent T-Test) |
| SN-MP (degrees) | 34.67 | 36.41 | 35.22 | 0.422 [‡] |
| | | | | (Independent T-Test) |
| **Presurgical orthodontic duration (days)** | | | | |
| Mean ± SD | 860.69 ± 514.768 | 0 | | |
| Min | 316 | - | | |
| Max | 1820 | - | | |

**Table 2.** The OQLQ score (total and subdomain) at each stage of the OFA group, compared using the Friedman and post hoc Dunn tests.

| Domain | T1 Mean ± S.D. | T2 Mean ± S.D. | T3 Mean ± S.D. | T4 Mean ± S.D. | T1–T2 *p* Value | T1–T3 *p* Value | T1–T4 *p* Value |
|---|---|---|---|---|---|---|---|
| Social (0–32) | 17.17 ± 7.634 | 11.87 ± 8.673 | 9.47 ± 6.653 | 8.03 ± 6.703 | 0.004 * | 0.000 * | 0.000 * |
| Esthetic (0–20) | 15.20 ± 3.800 | 10.10 ± 4.788 | 7.27 ± 4.362 | 6.70 ± 4.228 | 0.000 * | 0.000 * | 0.000 * |
| Function (0–20) | 12.17 ± 4.145 | 11.90 ± 4.759 | 9.33 ± 5.020 | 7.60 ± 4.484 | 1.000 | 0.031 * | 0.000 * |
| Awareness (0–16) | 8.97 ± 3.577 | 8.67 ± 3.809 | 7.30 ± 3.725 | 6.33 ± 3.863 | 1.000 | 0.594 | 0.036 * |
| Total (0–88) | 53.50 ± 15.965 | 42.53 ± 18.697 | 33.37 ± 17.582 | 28.60 ± 17.859 | 0.065 | 0.000 * | 0.000 * |

* Significant difference ($p < 0.05$).

The mean differences in the total OQLQ score and domain scores at each time point were compared between the groups (Table 4). At each time point (1 month, 3 months and 6 months after surgery), there was no significant difference in the change in the OQLQ score between the OFA and SFA groups.

**Table 3.** The OQLQ score (total and subdomain) at each stage of the SFA group, compared using the Friedman and post hoc Dunn tests.

| Domain | T1 Mean ± S.D. | T2 Mean ± S.D. | T3 Mean ± S.D. | T4 Mean ± S.D. | T1–T2 *p* Value | T1–T3 *p* Value | T1–T4 *p* Value |
|---|---|---|---|---|---|---|---|
| Social (0–32) | 16.43 ± 7.767 | 12.63 ± 7.721 | 9.70 ± 6.401 | 7.60 ± 6.414 | 0.242 | 0.000 * | 0.000 * |
| Esthetic (0–20) | 14.93 ± 3.095 | 10.13 ± 4.754 | 8.20 ± 4.766 | 5.97 ± 4.738 | 0.016 * | 0.000 * | 0.000 * |
| Function (0–20) | 11.77 ± 4.531 | 12.70 ± 4.388 | 9.33 ± 4.003 | 6.63 ± 4.263 | 1.000 | 0.167 | 0.001 * |
| Awareness (0–16) | 8.77 ± 4.199 | 8.17 ± 3.687 | 6.50 ± 3.812 | 5.67 ± 4.196 | 1.000 | 0.036 * | 0.001 * |
| Total (0–88) | 51.90 ± 15.273 | 43.30 ± 17.771 | 33.73 ± 15.739 | 25.87 ± 17.049 | 0.113 | 0.000 * | 0.000 * |

* Significant difference ($p < 0.05$).

**Table 4.** The mean differences at each time point, and comparisons between the OFA and SFA groups, using the Mann–Whitney U test.

| Domain | Δ T1–T2 OFA Mean Difference ± S.D. | SFA Mean Difference ± S.D. | *p* Value | Δ T1–T3 OFA Mean Difference ± S.D. | SFA Mean Difference ± S.D. | *p* Value | Δ T1–T4 OFA Mean Difference ± S.D. | SFA Mean Difference ± S.D. | *p* Value |
|---|---|---|---|---|---|---|---|---|---|
| Social | 3.4 ± 8.645 | 3.80 ± 7.332 | 0.976 | 5.80 ± 8.393 | 6.73 ± 7.821 | 0.859 | 7.23 ± 8.982 | 8.83 ± 8.284 | 0.604 |
| Esthetic | 4.20 ± 4.294 | 4.80 ± 4.715 | 0.504 | 7.033 ± 4.7306 | 6.733 ± 4.6899 | 0.657 | 7.60 ± 4.760 | 8.97 ± 3.908 | 0.229 |
| Function | 0.43 ± 5.649 | −0.93 ± 5.889 | 0.477 | 3.00 ± 5.452 | 2.43 ± 3.510 | 0.678 | 4.73 ± 4.299 | 5.80 ± 5.229 | 0.394 |
| Awareness | 0.20 ± 3.336 | 0.60 ± 3.793 | 0.656 | 1.57 ± 3.936 | 2.27 ± 4.168 | 0.528 | 2.53 ± 3.857 | 3.10 ± 3.772 | 0.381 |
| Total | 8.23 ± 15.156 | 8.60 ± 18.059 | 0.584 | 17.40 ± 17.704 | 18.17 ± 16.103 | 0.807 | 22.17 ± 18.688 | 26.03 ± 15.928 | 0.336 |

The domain effect sizes of the OFA and the SFA can be seen in Figure 1. Facial esthetics had the largest effect size, followed by social aspects of deformity, oral function, and awareness of facial deformities (E > S > F > A), which were the same in both groups.

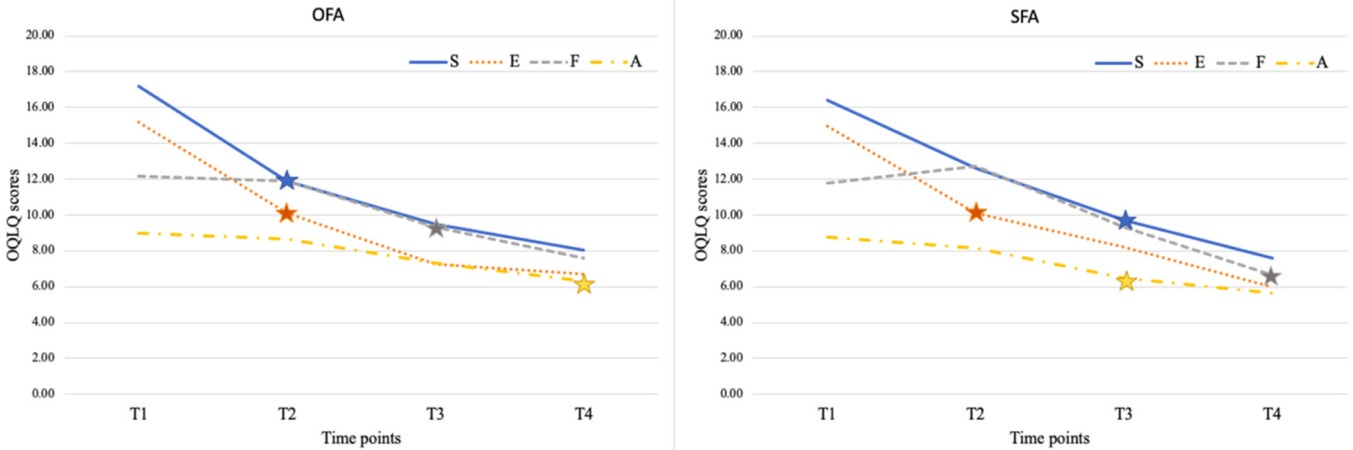

**Figure 1.** The change in domain scores at each time point.

## 4. Discussion

The general data and skeletal discrepancy between the OFA and SFA groups were compared, and showed no significant differences (Table 1). These findings were also related to a previous report, that the degree of skeletal discrepancy was the same in both groups [27]. The criteria for selecting between the SFA and the OFA were described by the orthodontic manner [10]. The presurgical orthodontic phase in the OFA took time, deteriorated occlusion and esthetics, caused discomfort or pain and affected patients' confidence [8,15–17]. The time of the presurgical orthodontic phase in the OFA was 860.69 ± 514.768 days in the current study. The SFA eliminated the presurgical orthodontic phase, decompensation started right after surgery, and promoted faster tooth movement rates by RAP [11], and previous studies showed significantly reduced treatment time [8,10].

At 1 month after surgery, the facial esthetics domain was improved in both the OFA and the SFA groups. This revealed that the first and foremost concern of Thai patients undergoing orthognathic surgery was esthetics. This result was confirmed by studies in Taiwan and the United Kingdom (UK) [23,28]. The results indicated that orthognathic surgery can significantly improve cosmetic outcomes, by immediately changing facial soft tissue [5]. In addition, the social aspects of the deformity domain were improved significantly only in the OFA group. The social aspects of deformity domain questions reflected the patients' thoughts about how other people regarded them and their appearance. The presurgical orthodontic decompensation phase in the OFA resulted in the patients' facial profiles deteriorating after such a long period [9,22]. When the skeleton was corrected immediately after surgery, the change in facial profile and esthetics improved dramatically: as a result, OFA patients could have experience of massive changes in the social aspects of

deformity within 1 month, while SFA patients required more time to adapt to socializing, from our clinical observation.

At 3 months after surgery, awareness of facial deformities was an additional domain in the SFA group that improved significantly. This domain's questions reflect awareness and concern about teeth, and the appearance of oneself and others. An improvement in this domain could mean that patients are more confident about noticing teeth, and their own facial appearance and that of others, as a result of esthetic improvement. Our clinical observation showed that the OFA patients had lower self-esteem and self-confidence during the presurgical orthodontic phase, and that this continued in the long term after surgery; the awareness of facial deformities improved later, at 6 months, in the OFA patients. The SFA patients, on the other hand, had no experience of the presurgical orthodontic phase, and their awareness of facial deformities improved at 3 months after surgery. Other studies have produced varying results in regard to the awareness of the facial deformities domain: a study in South Korea showed the same results [21], while the OFA and SFA groups in the UK study both demonstrated improvement after 6 weeks [23]; by contrast, a study in Taiwan found that, from 1 month to 12 months following surgery, there was no significant improvement in this domain in SFA patients [28].

Oral function was an additional domain in the OFA group that improved significantly. This delayed improvement of domain may have been a result of limitation of function at 1 month after surgery, when soft tissue swelling had not completely resolved, and early mobilization had just started. Previous research has shown that within a month of surgery, swelling significantly decreased, and that within 3 months, the healing was almost complete [29,30]; patients were motivated to eat a soft-to-regular diet during the second and third months after surgery; in addition, the OFA patients were in the finishing and detailing phase of orthodontic treatment, so that they could begin to function efficiently; therefore, 3 months after surgery, the oral function domain had significantly improved. A study in South Korea also showed that the oral function domain improved significantly after 3 months [21]. A study in the UK showed improvement after 6 weeks [23,28]. In the SFA, as no dental decompensation had been performed before surgery [7], leveling and aligning were performed after surgery. Patients could still experience discomfort as a result of orthodontic treatment, which could impede their ability to function. A study in Taiwan reported worsening of the oral function domain at 1 month, and improvement after 6 months [28]. Similar to the UK's result and our result, the oral function domain improved significantly at 6 months [23]. This may be concluded to be a result of improved occlusion in postsurgical orthodontic treatment. By contrast, the South Korea study reported that the oral function domain in both the OFA and the SFA improved after 3 months [21].

At 6 months after surgery, all domains and total scores improved significantly in both groups; likewise, previous studies have demonstrated that all domains in both the OFA and SFA groups improved significantly [21,23]. At 6 months, all postoperative sequelae had been resolved, bone healing had been completed, and occlusion was nearly completely corrected. The patients may also have become familiar with their new occlusion and appearance; therefore, quality of life improved significantly in all domains.

When comparing changes in scores across groups, the results revealed that there was no significant difference in the OQLQ score change in each domain, between the OFA and the SFA groups, at any stage: these results were similar to previous studies performed in other countries [15–17].

In addition, we also calculated the effect size, to compare the magnitude of change of each domain between groups and to other published studies. We found the same results in both the OFA and the SFA groups: facial esthetics had the largest effect size, followed by social aspects of deformity, oral function and awareness of facial deformities (E > S > F > A), representing the degree of concern in Thai patients from our center. In the same way, a study in South Korea showed the same effect size sequence of the OFA group as the SFA group: the sequences from high to low were facial esthetics, oral function, social aspects of deformity and awareness of facial deformities (E > F > S > A) [21]. In addition, another

study in the UK reported that the sequences of effect size in the OFA group were facial esthetics, social aspects of deformity, awareness of facial deformities and oral function (E > S > A > F), and that the sequences of effect size in the SFA group were facial esthetics, social aspects of deformity, oral function and awareness of facial deformities (E > S > F > A) [23]. From all of the above, it may be concluded that facial esthetics are the most impactful factor in the global patient's QoL after orthognathic surgery. The other domains differ, based on the method of approach and the region of study.

The limitation of this study was that randomization of the sample could not be done; however, the demographic data and level of skeletal discrepancy were compared, and were found not to be significantly different in reducing selection bias.

## 5. Conclusions

OFA and SFA showed QoL scores gradually improving in each domain until 6 months after surgery. The pattern of improvement in the SFA group was facial esthetics (E) at 1 month, followed by awareness of facial deformities (A) and social aspects of deformity (S) at 3 months, followed by oral function (F) at 6 months, respectively, after surgery. The pattern of improvement in the OFA group was facial esthetics (E) and social aspects of deformity (S) at 1 month, followed by oral function (F) at 3 months, followed by awareness of facial deformities (A) at 6 months, respectively, after surgery. The most concerning domain for Thai patients in our center's OQLQ was the facial esthetic domain. OQLQ is a useful questionnaire for follow up orthognathic surgery outcome.

**Author Contributions:** Conceptualization: S.P., A.K. and N.C.; investigation: C.T., S.P. and N.C.; formal analysis: C.T., S.P. and N.C.; manuscript preparation: C.T., A.K. and N.C. All authors have read and agreed to the published version of the manuscript.

**Funding:** The study was supported by Thammasat University Research Unit in Mineralized Tissue Reconstruction, Thammasat University, Thailand.

**Institutional Review Board Statement:** Ethical approval for the trial was obtained from The Human Research Ethics Committee of Thammasat University (Science) (project code 120/2563).

**Informed Consent Statement:** Written consent was obtained from each patient in advance of the study.

**Data Availability Statement:** Not applicable.

**Conflicts of Interest:** There are no conflict of interest to declare.

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
