# Peer review of "Comparison of “Orthodontic First” and “Surgery First” Approaches to Quality of Life in Orthognathic Surgery Patients: A Prospective Cohort Study"

_applsci, doi:10.3390/app122312137_

Round 1

Reviewer 1 Report

The study indicates similar results for both groups after months and only a slightly greater improvement in the SFA group, the advantage attenuated after 6 months. I believe that the study needed a control group for the accuracy of the results.

Reviewer 2 Report

The work is done appropriately additions are required regarding

1, the occlusal plane angulation and weather it has been changed or not.

2. Stability of the results considering the occlusion and post operative complications as well.

3. 2- weeks is too early for the patient to start occlusal to function normally 

Reviewer 3 Report

Dear Authors,

This paper addresses an interesting topic. 

However, I would recommend several modifications before considering its publication. Below these are some suggestions for You:

  1. Title:

    1. Include the type of your study

  2. Abstract:

    1. While OFA group was E, S -> F -> A at 1, 3, 6 months respectively after surgery” - please rewrite to make more readable, without abbreviations and arrows

  3. Material and methods: 

    1. Study design: not too much information or clearly presented; this paragraph should make an opportunity to repeat the study by someone else. 

    2. Eligibility criteria: where are inclusion criteria? Why Parkinson's disease etc.?  What about the patient's age? 

  4. Results:

    1. Figure 1. barely visible - resolution should be better

  5. Discussion: 

    1. I suggest to extent the limitations

  6. Conclusions

    1. Supported by results, but you should not use abbreviations - change it, please. Second, I do not understand arrows. 

  7. References:

    1. The article is well documented, but I suggest finding some new references.

Best regards and good luck

Round 2

Reviewer 1 Report

Please carefully check the way of expression and the bibliographic notes.

Reviewer 2 Report

Minor changes have been done 

Reviewer 3 Report

Dear Authors,

This is a revised version of your original manuscript. You have corrected it according to the suggestions and it can be accepted without any changes now. 

Best regards and good luck